# The inflammasome adaptor protein ASC promotes amyloid deposition in cryopyrin-associated periodic syndromes

Cristina Alarcón-Vila[1], Laura Hurtado-Navarro[1], Sandra V Mateo[1], Alejandro Peñín-Franch[1], Carlos M Martínez [1], Cristina Molina-López [1], María C Baños[1], Ana I Gómez[1], Javier Gómez-Román[2], Alberto Baroja-Mazo[1], Juan I Arostegui[3,4,5], Natalia Palmou-Fontana[6], Juan J Martínez-García [1,7✉] & Pablo Pelegrin [1,7✉]

Amyloid A (AA)-type amyloidosis is a severe complication associated with different monogenic autoinflammatory diseases (Lachmann et al, 2007). We present a patient diagnosed with late-onset cryopyrin-associated periodic syndromes (CAPS), who started with recurrent inflammatory manifestation at 51 years including fever, chills, urticaria-like rash, fatigue, myalgias, and nephrotic-range proteinuria (>12 mg/day). The patient was found to carry the myeloid-restricted c.924 A > T: p.Q308H (RefSeq NM_001243133.1) variant in the gene nucleotide oligomerization-domain, leucine-rich repeats, and pyrin domain-containing 3 (*NLRP3*), with a variant allele frequency (VAF) in peripheral blood of 5.1%, compatible with gene mosaicism (Mensa-Vilaró et al, 2019). Clinical manifestations were effectively managed with anti-IL-1 therapy (Fig. 1A). In addition, the patient experienced faecal incontinence due to intestinal AA-type amyloidosis (Fig. 1B) and had elevated levels of serum amyloid A protein (SAA), which were normalized during anti-IL-1 treatment (Fig. 1C). Since the p.Q308H NLRP3 variant has been reported in a limited number of patients as a "likely pathogenic" variant (Mensa-Vilaró et al, 2019), with no functional studies performed, herein we aimed to characterize its consequences on the NLRP3-inflammasome activity. For this proposal, we ectopically expressed this variant in HEK293T cells and observed a significantly increased puncta distribution of mutant NLRP3 compared to wild-type NLRP3 expression (Figs. 1D and EV1A). This puncta distribution have been previously found for other CAPS-associated NLRP3 variants and during NLRP3-inflammasome activation (Baroja-Mazo et al, 2014). In addition, co-expression of the inflammasome adaptor protein apoptosis-associated speck-like protein containing a caspase activation domain (ASC) with mutant p.Q308H NLRP3 resulted in a higher number of cells with ASC oligomers (Figs. 1E and EV1B). These evidences confirm that the p.Q308H NLRP3 variant promotes the formation of active NLRP3-inflammasomes and support for a gain-of-function behaviour. This aligns with CAPS pathophysiology, which is characterized by constitutive activation of the NLRP3-inflammasome as a consequence of gain-of-function *NLRP3* variants (Molina-López et al, 2024). The activation of the NLRP3-inflammasome in CAPS also leads to the release of large inflammasome oligomers, primarily composed of ASC oligomeric filaments, which have been identified extracellularly during acute inflammatory flares in CAPS patients (Baroja-Mazo et al, 2014). We have previously found ASC oligomers in the plasma of different late-onset CAPS with myeloid-restricted NLRP3 variants and developing AA-type amyloidosis (Rowczenio et al, 2017). In addition, mouse models of CAPS exhibit tissue amyloid deposition (Bertoni et al, 2020), and the macrophages are responsible for the inflammatory phenotype (Frising et al, 2022). Recent studies have demonstrated that extracellular deposition of circulating ASC oligomers in tissues can act as seeds for amyloid deposition, which may finally lead to clinical AA-type amyloidosis (Losa et al, 2024; Venegas et al, 2017). We observed the presence of ASC alongside amyloid foci in the intestine of the late-onset CAPS patient here described (Fig. 1B), similar to the observation that ASC expression correlated with amyloidosis in familial Mediterranean fever patients (Balci-Peyrircioglu et al, 2008). Since the late-onset CAPS patient here reported carried a myeloid-restricted *NLRP3* gain-of-function variant, it is tentative to hypothesize that ASC oligomers could be released from myeloid cells in CAPS. To confirm this hypothesis, we analysed ASC oligomerization in different leukocyte subpopulations from CAPS patients carrying different germline *NLRP3* gain-of-function variants. After 4 h of in vitro culture we observed that monocytes, but not neutrophils or lymphocytes, were producing ASC oligomerization (Fig. 1F), albeit neutrophils were expressing ASC at higher levels than lymphocytes, but less than monocytes (Fig. EV1C), indicating that monocytes will be the main source of extracellular ASC oligomers in CAPS. To confirm a role of ASC in AA-type amyloid deposition, we used a mouse model of AA-type amyloidosis by administering casein subcutaneously daily over 25 days. This treatment led to amyloid

[1]Molecular Inflammation Group, Biomedical Research Institute of Murcia (IMIB), Murcia, Spain. [2]Anatomical Pathology Service, Marqués de Valdecilla University Hospital-IDIVAL, University of Cantabria, Santander, Spain. [3]Department of Immunology, Hospital Clínic, Barcelona, Spain. [4]Biomedical Research Institute August Pi i Sunyer, Barcelona, Spain. [5]School of Medicine, University of Barcelona, Barcelona, Spain. [6]Rheumatology Service, Immunopathology Group, Marqués de Valdecilla University Hospital-IDIVAL, Santander, Spain. [7]Department of Biochemistry and Molecular Biology B and Immunology, Faculty of Medicine, University of Murcia, Murcia, Spain.
✉E-mail: juanjose.martinez20@um.es; pablopel@um.es
https://doi.org/10.1038/s44321-024-00176-1 | Published online: 5 December 2024

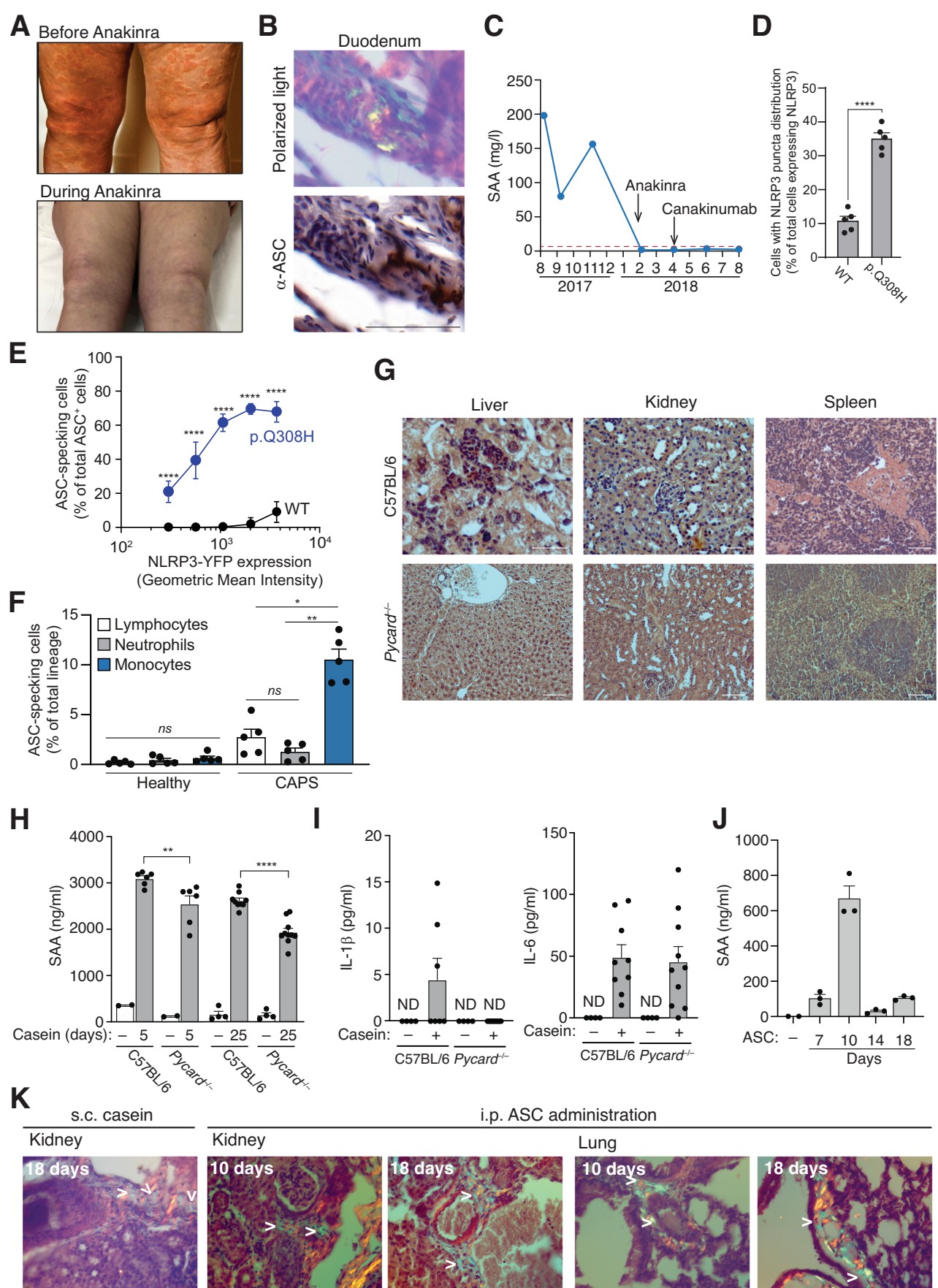

◄ **Figure 1.** Amyloid deposition correlates with extracellular ASC in CAPS.

(A) Cutaneous manifestations of a patient with late-onset cryopyrin-associated periodic syndrome (CAPS) with myeloid-restricted gain-of-function *NLRP3* somatic mutation p.Q308H before and after Anakinra treatment. (B) Congo-red and ASC-stained duodenum from the CAPS patient with amyloidosis. (C) Serum amyloid A (SAA) concentration in the plasma of the CAPS patient before and after IL-1 blocking therapy, red dotted line denotes 6 mg/l which are the reference value for a healthy adult. (D) Percentage of HEK293T cells with puncta distribution of NLRP3 wild type (WT) or p.Q308H; Centre values represent the mean ($n = 5$ independent experiments) and error bars represent s.e.m. Unpaired *t*-test, two tails (****$p < 0.0001$). (E) Percentage of HEK293T cells with ASC oligomers after expression of different amounts of NLRP3 WT ($n = 5$ independent experiments) or p.Q308H ($n = 5$ independent experiments); Centre values represent the mean and error bars represent s.e.m. Two-way ANOVA (****$p < 0.0001$). (F) Percentage of ASC specking cells in whole blood cultured for 4 h gating on lymphocytes (white bar), neutrophils (grey bar) and monocytes (blue bar) from healthy donors and germline CAPS patients with the NLRP3 variants p.R262W, p.D305N, p.T350M and p.A441T; Centre values represent the mean ($n = 5$ healthy, $n = 5$ patients) and error bars represent s.e.m.; one-way ANOVA (*$p = 0.0270$; **$p = 0.0061$; ns: not significant $p > 0.05$). (G) Congo-red stained in liver, kidney and spleen sections from C57BL/6 and *Pycard$^{-/-}$* mice after 25 days of casein administration. (H, I) Concentration of SAA (H), IL-1β and IL-6 (I) in the serum of mice treated as in (F) after 5 and 25 days (H) or 25 days (I); for (H), Two-way ANOVA (**$p = 0.0069$; ****$p < 0.0001$) with saline: $n = 2$ mice day 2, $n = 4$ mice day 25; casein: $n = 6$ mice day 5, $n = 9$ mice WT day 25, $n = 10$ mice *Pycard$^{-/-}$* day 25; for (I), IL-6, saline: $n = 4$ mice, casein: $n = 9$ mice WT, $n = 10$ mice *Pycard$^{-/-}$*; for (I), IL-1β $n = 3$ mice; Centre values represent the mean and error bars represent s.e.m. (J) Concentration of SAA in serum of *Pycard$^{-/-}$* mice after 7, 10, 14, or 18 days of intraperitoneal injection of ASC oligomers; $n = 2$ mice for control, $n = 3$ mice for the different days; Centre values represent the mean and error bars represent s.e.m. (K) Congo-red stained in kidney and lung sections from *Pycard$^{-/-}$* mice after 10 and 18 days of intraperitoneal ASC oligomers administration (as control, kidney sections from the casein model presented in (G) is shown). Arrowheads denote areas of amyloid deposition. Scale bars: 50 μm. Source data are available online for this figure.

deposition in the liver, kidney, and spleen (Figs. 1G and EV2), along with a corresponding increase in serum SAA concentration (Fig. 1H). ASC-deficient (*Pycard$^{-/-}$*) mice showed a reduction in amyloid deposition in tissues (Figs. 1G and EV2), despite serum SAA concentration remained elevated (Fig. 1H). Notably, when comparing SAA concentrations in the serum of casein-treated wild-type mice to *Pycard$^{-/-}$* mice after 25 days, SAA concentration was significantly lower in *Pycard$^{-/-}$* (Fig. 1H). As expected, *Pycard$^{-/-}$* mice exhibited decreased production of IL-1β but maintained normal levels of IL-6 following casein administration (Fig. 1I). In wild-type mice, casein induces steatohepatitis and significant inflammatory infiltration in the liver, which was absent in *Pycard$^{-/-}$* mice (Figs. 1G and EV2). To confirm that extracellular ASC oligomeric particles promote amyloid deposition in CAPS, we intraperitoneally injected ASC oligomers produced with p.D305N NLRP3 seeds into *Pycard$^{-/-}$* mice. ASC oligomers administration resulted in increased SAA concentrations in the serum (Fig. 1J). These mice also showed presence of amyloid deposition in the kidneys and lungs (Figs. 1K and EV3). Overall, NLRP3-inflammasome activation in CAPS is associated with inflammation, elevated serum SAA concentration, and increase of monocyte-dependent extracellular ASC oligomers that promote amyloid deposition in tissues, leading to AA-type amyloidosis. Recently, various nanobodies have been developed to target extracellular ASC (Bertheloot et al, 2022; Losa et al, 2024), offering a potential therapy to prevent AA-type amyloidosis development in CAPS and potentially in other chronic inflammatory diseases.

## Methods

### Reagents and tools table

| Reagent/Resource | Reference or Source | Identifier or Catalog Number |
|---|---|---|
| **Experimental models** | | |
| C57BL/6J mice (*M. musculus*) | Jackson Laboratories | RRID: IMSR_JAX:000664 |
| ASC deficient mice in C57BL/6J background (*Pycard$^{-/-}$*) | Mariathasan et al, 2004 | N/A |
| HEK293T cells (*H. sapiens*) | American Type Culture Collection | CRL-11268 |
| HEK293T cells (*H. sapiens*) stably expressing human NLRP3 p.D303N | Baroja-Mazo et al, 2014 | N/A |
| Human primary whole blood | This study | N/A |
| **Recombinant DNA** | | |
| NLRP3 wild-type tagged with Yellow Fluorescent Protein (YFP) | Baroja-Mazo et al, 2014 | N/A |
| ASC tagged with Red Fluorescent Protein (RFP) | Franklin et al, 2014 | N/A |
| ASC tagged with Yellow Fluorescent Protein (YFP) | Baroja-Mazo et al, 2014 | N/A |
| NLRP3 p.Q308H tagged with Yellow Fluorescent Protein (YFP) | This study | N/A |
| **Antibodies** | | |
| PE anti-ASC (TMS-1) Antibody | Biolegend | Cat. 653902 (clone HASC-71) |
| Vector ImmPress anti-mouse | Vector Laboratories | Cat. MP-7402 |

| Reagent/Resource | Reference or Source | Identifier or Catalog Number |
|---|---|---|
| FITC Mouse Anti-Human CD14 | BD Biosciences | Cat. 557153 (clone M5E2) |
| APC Mouse Anti-Human CD15 | BD Biosciences | Cat. 551376 (clone HI98) |
| **Oligonucleotides and other sequence-based reagents** | | |
| pcDNA3.1/V5-His Topo | Life Technologies | Cat. K480001 |
| **Chemicals, Enzymes and other reagents** | | |
| Mouse IL-1 beta ELISA | Thermo Fisher Scientific | Cat. 88-7013-88; RRID:AB_2574946 |
| Mouse IL-6 ELISA Kit - Quantikine | R&D Systems | Cat. M6000B-1; RRID: AB_2877063 |
| Mouse SAA/SAA1 ELISA Kit | Biorbyt | Cat. ORB170167 |
| MycoProbe Mycoplasma Detection Kit | R&D Systems | Cat. CUL001B |
| ASC oligomers | This study | N/A |
| Penicillin-Streptomycin | Corning | Cat. 30-002-CI |
| Dulbecco's modified Eagle's medium (DMEM-F12) w/o L-Glutamine, w/o Hepes Sterile Filtered | Biowest | Cat. L0090-500 |
| RPMI-1640 Medium liquid without L-glutamine | Sigma | Cat. R0883 |
| L-Glutamine | Gibco | Cat. 25030-032 |
| Fetal Bovine Serum (FBS) | Biowest | Cat. S181A |
| OptiMEM reduced Serum Media | Gibco | Cat. 51985-026 |
| GlutaMAX | Thermo Fisher Scientific | Cat. 35050038 |
| Lipofectamine 2000 | Invitrogen | Cat. 11668-019 |
| Vitamin-free casein | Merck Millipore | Cat. C6554 |
| HEPES | Sigma-Aldrich | Cat. H3375 |
| Sucrose | Sigma-Aldrich | Cat. S7903 |
| KCl | Panreac | Cat. 141494 |
| $MgCl_2$ | Sigma-Aldrich | Cat. M2670 |
| EDTA | Sigma-Aldrich | Cat. E5134 |
| EGTA | Sigma-Aldrich | Cat. E4378 |
| CHAPS | Calbiochem | Cat. 220201 |
| Percoll | Merck Millipore | Cat. P4937 |
| Mayer's haematoxylin | Panreac AppliChem | Cat. 254766.1211 |
| Alcoholic eosin Y | Panreac AppliChem | Cat. 251301.1211 |
| Target Retrieval Solution, High pH | Agilent-Dako | Cat. GV80411-2 |
| DAB+ Substrate Chromogen System | Agilent-Dako | Cat. GV82511-2 |
| Congo Red Stain Kit/Amyloid Stain Kit | Abcam | Cat. ab150663 |
| **Software** | | |
| GraphPad Prism version 9 | GraphPad Software Inc | www.graphpad.com |
| FlowJo™ v10.8 Software | BD Life Sciences | www.bdbiosciences.com/en-us/products/software/flowjo-v10-software |
| NIS Elements | Nikon | www.microscope.healthcare.nikon.com/products/software/nis-elements |
| Zen Ver. 3.0 Blue Edition | Karl Zeiss | www.zeiss.com/microscopy/en/products/software/zeiss-zen.html |
| **Other** | | |
| Nikon Eclipse Ti | Nikon | N/A |
| Zeiss Axio Scope A10 | Karl Zeiss | N/A |
| LSRFortessa | BD Biosciences | N/A |
| FACS Canto | BD Biosciences | N/A |
| Synergy Mx Plate Reader | BioTek | N/A |

## Preparation of experimental models and subject details

### CAPS patient

Written informed consent was obtained from the human volunteers involved in this study to use their biological samples and clinical data for this study. Human samples were used following standard operating procedures with appropriate approval of the Ethical Committee of the Clinical University Hospital Virgen de la Arrixaca (Murcia, Spain) and the principles of the WMA Declaration of Helsinki and the Department of Health and Human Services Belmont Report.

We describe a patient in their 60 s (2017) of Spanish descent, born from non-consanguineous parents, with no family history of CAPS. The medical history of the patient includes uterine cancer with hysterectomy in 1995, bilateral neurosensorial hearing loss and papilledema diagnosed in 2010, and ankylosing spondylitis diagnosed in 2014. In 2015, she was diagnosed with faecal incontinence without identifiable triggers. The symptoms included low-grade fever, conjunctivitis, polyarthralgia, and oligoarthritis affecting the knees, wrists, elbows, and ankles. Elevated serum amyloid protein levels were detected (Fig. 1C), and AA-type amyloidosis were found in the duodenum (Fig. 1B) and rectum, but not in the oesophagus or ileum. Initial treatment began in 2014 with various anti-TNF biologics (Adalimumab, Golimumab, and Certolizumab), with the latter showing limited improvement in dermatological lesions and alleviating inflammatory back pain. Anti-TNF treatment was discontinued in 2017, and Anakinra was initiated, resulting in a partial response. In 2018, Anakinra was replaced with Canakinumab (300 mg every 15 days), which led to a dramatic positive response and good control of clinical symptoms. A myeloid-restricted *NLRP3* variant in exon 3 (c.924 A > T; p.Q308H; RefSeq NM_001243133.1) was identified in mosaicism (Mensa-Vilaró et al, 2019), leading to a diagnosis of late-onset Muckle-Wells syndrome, the intermediate severity among CAPS phenotypes. Treatment with anti-IL-1 drugs was maintained every 15 days. However, during the COVID-19 pandemic, the anti-IL-1 treatment was interrupted, and the patient developed progressive systemic AA-type amyloidosis, severe diarrhoea, and renal failure, which led to a kidney transplantation. Post-transplantation, anti-IL-1 treatment was resumed, but the clinical symptoms did not improve. Switching to Tocilizumab also failed to alleviate the symptoms. Canakinumab was then administered every 28 days, which improved the diarrhoea and skin urticarial-like rash. In 2022, the patient suffered a stroke due to hypertension, resulting in cognitive impairment. In June 2024, the patient passed away due to sepsis and multiorgan failure.

In this study we also analysed blood samples collected in EDTA anticoagulated tubes from healthy donors ($n = 5$) and CAPS (Muckle-Wells syndrome) patients who carried the following germline NLRP3 gain-of-function variants: p.R262W ($n = 1$), p.D305N ($n = 1$), p.T350M ($n = 1$) and p.A441T ($n = 2$). By the time the blood was extracted, all CAPS patients were under IL-1 blocking therapy with inactive disease.

### Cell culture and transfections

Human blood (50 µl) was cultured in polystyrene flow cytometry tubes (Falcon) with RPMI 1640 medium (Sigma) containing 10% FCS and 2 mM Glutamax (Thermo Fisher) for 4 h at 37 °C in a humidified 5% $CO_2$ incubator.

HEK293T cells (CRL-11268, American Type Culture Collection) were maintained in Dulbecco's modified Eagle's medium (DMEM)/F-12 (1:1) (Biowest) supplemented with 10% fetal calf serum (FCS) (Biowest) and 2 mM L-glutamin (Gibco). Cells were maintained at 37 °C in a humidified 5% $CO_2$ incubator. Cell line was not authenticated but was free of mycoplasma by routinely testing with the MycoProbe Mycoplasma Detection Kit following manufacturer instructions (R&D Systems). Lipofectamine 2000 (Invitrogen) was used according to the manufacturer's instructions for the transfection of HEK293T cells using 0.1 to 1 µg of NLRP3 wild-type or p.Q308H construct, and with 0.1 µg of ASC tagged with Red Fluorescent Protein (RFP). Microscopy assays were performed after 24 h of transfection.

### Animals

All experimental protocols for animal handling were refined and approved by the ethics committee of the University of Murcia (reference CEEA 554/2019), the Biosecurity committee of the University of Murcia (reference CBE217/2019) and the Animal Health Service of the General Directorate of Fishing and Farming of the Council of Murcia (*Servicio de Sanidad Animal, Dirección General de Ganadería y Pesca, Consejería de Agricultura y Agua Región de Murcia*, reference A13211201). C57BL/6 mice (WT, wild-type, RRID: IMSR_JAX:000664) were obtained from the Jackson Laboratories, and ASC deficient mice in C57BL/6 background (*Pycard⁻/⁻*) (Baroja-Mazo et al, 2014) was a generous gift from I. Couillin (University of Orleans and CNRS, Orleans, France). For all experiments, male and female mice between 8 and 10 weeks of age were used, no differences were observed in SAA concentrations among males and females. Mice were bred in specific pathogen-free conditions with a 12:12 h light-dark cycle and used in accordance with the Spanish national (Royal Decree 1201/2005 and Law 32/2007) and EU (86/609/EEC and 2010/63/EU) legislation. Sample size was calculated estimating an alpha of 0.05 and a statistical power of 94%. Animals were randomized before starting procedures and initial weight of animals was similar among randomized groups. Subcutaneous daily injection of 500 µl of saline sterile solution or 10% vitamin-free casein (Merck) in 0.05 M of $NaHCO_3$ for 5 or 28 days. In another set of experiments, recombinant ASC oligomers were intraperitoneal injected at a dose of $4 \times 10^6$ for mice. Mice were sacrificed at days 7, 10, 14 and 18 after injection. After procedures, the animals were euthanized by $CO_2$ inhalation and peritoneal lavages and blood and tissue samples were collected. Blood samples were obtained by thoracic aorta and were centrifuged at $12,500 \times g$ for 10 min. The recovered serum was stored at −80 °C until further use. For tissue harvesting the abdominal wall was exposed, the organs were removed using scissors and forceps and were fixed and paraffin-embedded or stored at −80 °C for future analysis.

### Plasmid constructs

The mutation of human NLRP3 (UniProt #Q96P20) was generated by overlapping polymerase chain reaction (PCR) to introduce a point mutation (p.Q308H) and cloned into pcDNA3.1/V5-His Topo (Life Technologies). Sequencing of the construct was performed to confirm correct modification and the absence of unwanted mutations. The construct was double-tagged using overlapping PCR in the N terminus to YFP. On the other hand, human ASC (Uniprot #Q9ULZ3) expression vector tagged with RFP at the C terminus was

kindly provided by E. Latz (Institute of Innate Immunity, University Hospital Bonn, Bonn, Germany).

### ASC oligomer production

We followed the protocol registered at https://doi.org/10.21769/BioProtoc.1480 with modifications. In particular, HEK293T cells stably expressing human NLRP3 p.D305N were transiently transfected with an expression vector coding for human ASC-YFP tagged at C-terminal using Lipofectamine™ 2000 according to the manufacturer instructions. After 48 h of transfection, ASC oligomerization was confirmed by fluorescence microscope (Nikon Eclipse Ti) and cells were resuspended on 600 µL Buffer A composed of 20 mM HEPES (Sigma-Aldrich), 300 mM Sucrose (Sigma-Aldrich), 10 mM KCl (Panreac), 1.5 mM MgCl₂ (Sigma-Aldrich), 1 mM EDTA (Sigma-Aldrich), 1 mM EGTA (Sigma-Aldrich). Cell suspension was mechanically lysed by pressuring 10 times with 1 ml syringe and 20 G needle followed by pressuring 20 times with 1 ml syringe and 25 G needle. Mechanical cell lysis was accompanied by a final heat shock cell lysis with fast freezing on liquid $N_2$, followed by 37 °C water bath unfreezing. Lysates were centrifuged at $400 \times g$ for 8 min at 4 °C. Resulting supernatants were mixed with 2 volumes of 2x CHAPS buffer composed of Buffer A with 0.2% CHAPS (Calbiochem), and then centrifuged at $2000 \times g$ for 10 min at 4 °C. Filtered volume was mixed 1:1 with 2x CHAPS buffer and centrifuged at $2300 \times g$ for 8 min at 4 °C. Pellet was washed by resuspending it in 1 mL 1x CHAPS buffer followed by centrifugation at $5000 \times g$ for 8 min at 4 °C. This step was repeated 3 times. Pellet was added into a 40% Percoll (Merck Millipore) diluted in 1x CHAPS, and centrifuged at $16,000 \times g$ for 10 min at 4 °C. After centrifugation, the intermedial phase was collected and washed with 1 mL 1x CHAPS buffer and centrifuged at $16,000 \times g$ for 10 min at 4 °C. Pellet was resuspended in 100 mL 1x CHAPS buffer and obtained ASC oligomers were counted by fluorescence microscope (Nikon Eclipse Ti). ASC oligomers were diluted into 1x PBS at a concentration of $4 \times 10^6$ oligomers per 100 µl, ready to use for in vivo experiments.

### Fluorescence microscopy

To analyse wild-type or p.Q308H NLRP3 puncta distribution, HEK293T cells were transfected as described before. Images were acquired with a Nikon Eclipse Ti microscope equipped with a 20× S Plan Fluor objective (numerical aperture, 0.45) and a digital Sight DS-QiMc camera (Nikon) and 472 nm/520 nm and 543 nm/593 nm filter sets (Semrock), and the NIS-Elements AR software (Nikon). An example of puncta distribution in NLRP3 expressing cells is shown in Fig. EV1A. Images were analysed with ImageJ (US National Institutes of Health).

### Flow cytometry

Intracellular ASC-RFP-speck formation in HEK293T cells was evaluated after 24 h post-transfection by Time of Flight for Inflammasome Evaluation (TOFIE) in different gates with increasing mean fluorescence intensity for NLRP3-YFP (Fig. EV1B). Samples were analysed by flow cytometry using LSRFortessa (BD Biosciences) and FlowJo™ v10.8 Software (BD Life Sciences).

Intracellular ASC oligomerization in blood human leukocytes was evaluated by TOFIE flow cytometry technique. Cultured human whole blood cells were stained using the phycoerythrin (PE) conjugated mouse monoclonal anti-ASC antibody (clone HASC-71, catalogue 653903, Biolegend, 1:500). Monocytes were stained and gated with fluorescein (FITC)-conjugated mouse anti-CD14 monoclonal antibody (clone M5E2; catalogue #557153; BD Biosciences, 1:10) and neutrophils were gated in a forward scatter/side scatter (FSC/SSC) dot plot and gated with allophycocyanin (APC)-conjugated mouse anti-CD15 monoclonal antibody (clone HI98, catalogue 551376, BD Biosciences, 1:10). Lymphocytes were gated in a FSC/SSC dot plot. Human blood samples were analysed by flow cytometry using FACS Canto (BD Biosciences) and the FlowJo™ v10.8 Software (BD Life Sciences).

### Histology

To determine presence of ASC expression associated with amyloid deposits within the affected tissue, samples from human intestine (duodenum) were obtained by endoscopic biopsy, formalin-fixed and paraffin-embedded. 4-µm-thick sections stained with a standard haematoxylin and eosin (H&E) technique. Briefly, after deparaffination and rehydration, sections were stained with Mayer's haematoxylin (Thermo Scientific) and alcoholic eosin Y (Thermo Scientific). To study co-localization of ASC and amyloid deposits, a combined indirect immunohistochemical procedure and a Congo red staining were performed in 10-µm-thick sections. For immunohistochemistry, after deparaffination and rehydration, a heat-induced antigen demasking procedure was carried out using a commercial kit (High pH target retrieval solution, Agilent-Dako). After the endogenous peroxidase inactivation and background blockage, the sections were incubated at 4 °C overnight with the primary anti-ASC antibody (Biolegend mouse anti-ASC clone HASC-71; ref: 653902; dilution 1:250). Sections were then incubated with the secondary-HRP labelled polymer (Vector ImmPress anti-mouse, Vector Laboratories, Burlingame, USA) for 30 min at 37 °C. Immunoreaction was finally revealed with 3-3′ diaminobencidine (DAB) for 5 min at room temperature by using a commercial kit (DAB chromogen kit, Agilent-Dako). Positive immunoreaction was identified as a dark-brown precipitated. After the immunohistochemical procedure, a Congo red stain was performed on the sections by using a commercial kit (Congo red stain, Abcam) following the manufacturer's instructions. The sections were finally contrasted with Mayer's haematoxylin, dehydrated, and mounted in permanent media. Sections from human kidney inflammatory amyloidosis were complementary used as a positive control. Immunohistochemical staining was evaluated by using a brightfield standard microscope (Zeiss Axio Scope A10, Karl Zeiss). To evaluate the presence of amyloid deposits the sections were visualized under polarized light by using a polarization module on the same microscope. The amyloid protein was identified as light-green extracellular deposits. Representative images were obtained with a high-resolution digital camera (AxioCam 506 color, Karl Zeiss), by using a specialized software (Zeiss Zen Ver. 3.0 Blue Edition). Pathologist was blinded to sample identity.

### ELISA

Mouse serum was used with the following ELISA kits according to manufacturer instructions: mouse IL-1β (RRID: AB_2574946, Thermo Fisher Scientific), mouse IL-6 (RRID: AB_2877063, R&D Systems), and mouse SAA (ORB170167, Biorbyt). ELISA was read in a Synergy Mx (BioTek) plate reader at 450 nm and corrected at 540 nm.

## Statistical analysis

Statistical analyses were performed using GraphPad Prism version 9 (GraphPad Software Inc). Normality of the samples was determined with D'Angostino and Pearson omnibus K2 normality test. Outliers from data sets were identified by the ROUT method with $Q = 1\%$ and were eliminated from the analysis and representation. All data are shown as mean values and error bars represent standard error (SEM). Unpaired *t*-test, two tails, was used to compare two groups, for comparison of multiple groups a two-way ANOVA was used. Significance *p* values are either indicated by exact number in the figure legend and using the following symbols in the figure: ****$p < 0.0001$; **$p < 0.01$.

## Peer review information

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

## Acknowledgements

We thank E Latz (Institute of Innate Immunity, University Hospital Bonn, Bonn, Germany) for the human ASC expression vector tagged with RFP and I Couillin (University of Orleans and CNRS, Orleans, France) for ASC deficient mice. We thank the pathology service or the Biomedical Research Institute of Murcia (Spain), the central core facility of cell culture of the University of Murcia (Spain) for flow cytometry facilities, and animal facilities at University of Murcia and the Biomedical Research Institute of Murcia (Spain). This work was supported by grants from MCIN/AEI/10.13039/501100011033 and European Union «Next Generation EU/PRTR» (grants PID2020-116709RB-I00, CNS2022-135101, PID2023-147531OB-I00 and RED2022-134511-T to PP), FEDER/Ministerio de Ciencia, Innovación y Universidades – Agencia Estatal de Investigación (grant SAF2017-88276-R to PP), Fundación Séneca (grant 21897/PI/22 to PP, grant 21967/JLI/22 to JJMG), and the Instituto Salud Carlos III (grant AC22/00009 to PP; PI20/00185 to ABM). LH-N was supported by the fellowship 21214/FPI/19 (Fundación Séneca, Región de Murcia, Spain). CM-L was funded by the fellowship PRE2018-086824 (Ministerio economía y competitividad). SVM was supported by a fellowship from Instituto Salud Carlos III (FI21/00073). JJGM was supported by a Maria Zambrano fellowship at University of Murcia.

## Author contributions

**Cristina Alarcón-Vila**: Conceptualization; Formal analysis; Investigation; Methodology. **Laura Hurtado-Navarro**: Formal analysis; Investigation; Writing—review and editing. **Sandra V Mateo**: Formal analysis; Investigation. **Alejandro Peñín-Franch**: Investigation; Methodology. **Carlos M Martínez**: Formal analysis; Investigation; Methodology; Writing—review and editing. **Cristina Molina-López**: Formal analysis; Investigation; Writing—review and editing. **Maria C Baños**: Investigation. **Ana I Gómez**: Investigation. **Javier Gómez-Román**: Resources; Writing—review and editing. **Alberto Baroja-Mazo**: Conceptualization; Resources; Funding acquisition; Writing—review and editing. **Juan I Arostegui**: Conceptualization; Resources; Supervision; Writing—review and editing. **Natalia Palmou-Fontana**: Resources; Formal analysis; Writing—original draft; Writing—review and editing. **Juan J Martínez-García**: Conceptualization; Investigation; Writing—original draft; Writing—review and editing. **Pablo Pelegrin**: Conceptualization; Resources; Supervision; Funding acquisition; Visualization; Writing—original draft; Project administration; Writing—review and editing.

## Disclosure and competing interests statement

PP and JJMG declare that they are inventors in a patent filed on March 2020 by the *Fundación para la Formación e Investigación Sanitaria de la Región de*

*Murcia* (PCT/EP2020/056729) for a method to identify NLRP3-immunocompromised sepsis patients. PP, LH-N, JJMG and ABM are co-founders of Viva in vitro diagnostics SL, but declare that the research was conducted in the absence of any commercial or financial relationships that could be construed as a potential conflict of interest. The remaining authors declare no competing interests.

# Expanded View Figures

**Figure EV1.   NLRP3 p.Q308H resulted in puncta distribution and ASC oligomerization.**

(**A**) Representative images of HEK293T cells transfected with NLRP3 wild type (WT) or p.Q308H tagged with YFP; Arrowheads denote cells with a puncta distribution of NLRP3. Scale bars: 10 μm. (**B**) Gating strategy to analyse the percentage of ASC specking cells in five different gates with low constant ASC-RFP expression and increased expression of NLRP3-YFP wild type (top) or p.Q308H (bottom) calculated as mean fluorescence intensity (MFI). NLRP3-YFP MFI corresponding to the five gates are: 1: 300.25; 2: 564; 3: 1063.25; 4: 2023.75; 5: 3693. As example, in the right panels, the percentage of ASC specking cells is shown in the gate number three for NLRP3 expression. (**C**) ASC MFI in human blood lymphocytes (white bar), neutrophils (grey bar) and monocytes (blue bar) cultured for 4 h, from healthy donors and germline CAPS patients whole blood with the NLRP3 variants p.R262W, p.D305N, p.T350M and p.A441T; Centre values represent the mean ($n = 5$ healthy, $n = 5$ patients) and error bars represent s.e.m; one-way ANOVA ($^{\#\#}p = 0.0019$; $^{**}p = 0.0080$; $^{++}p = 0.0040$; *ns*: not significant $p > 0.05$).

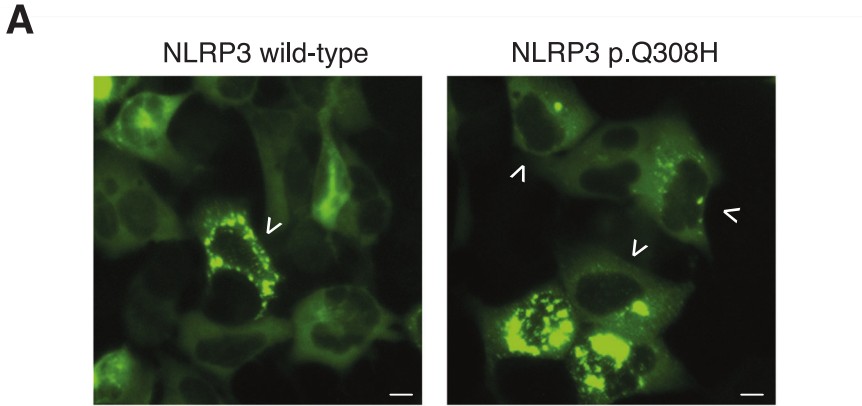

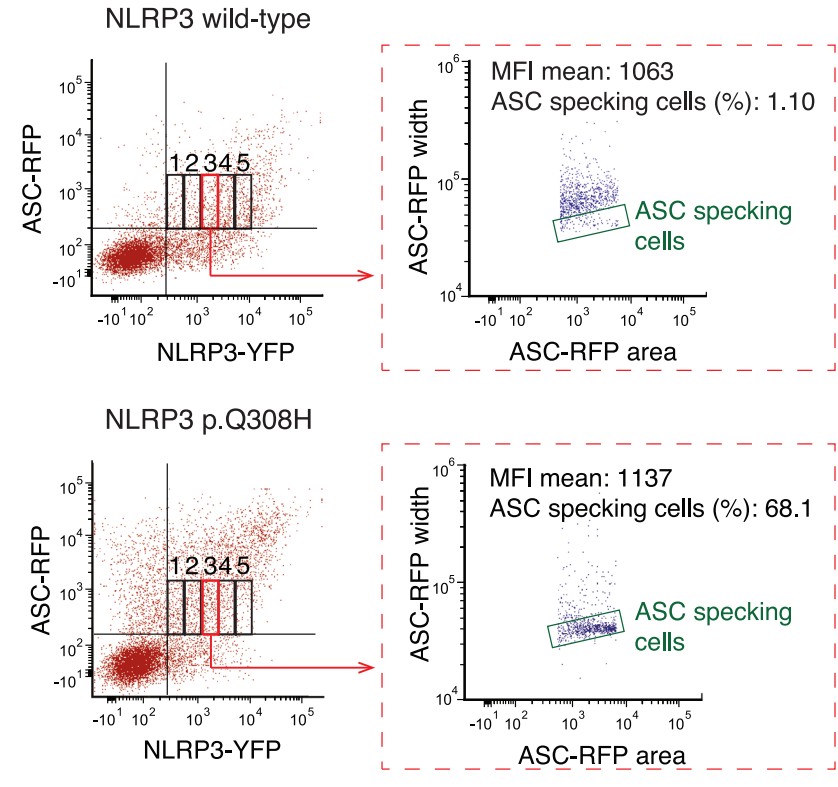

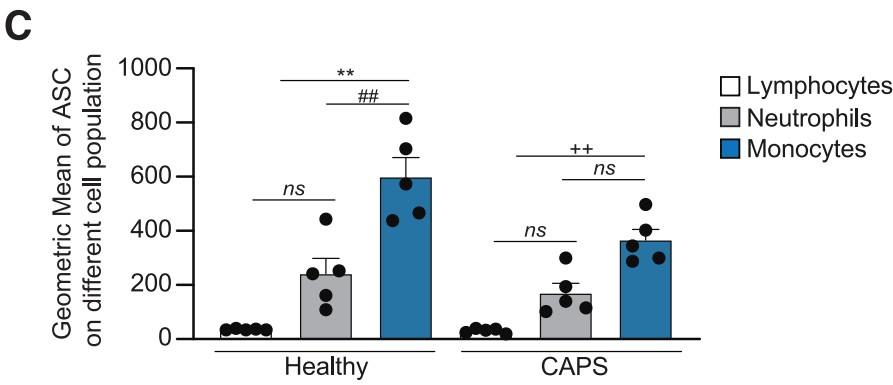

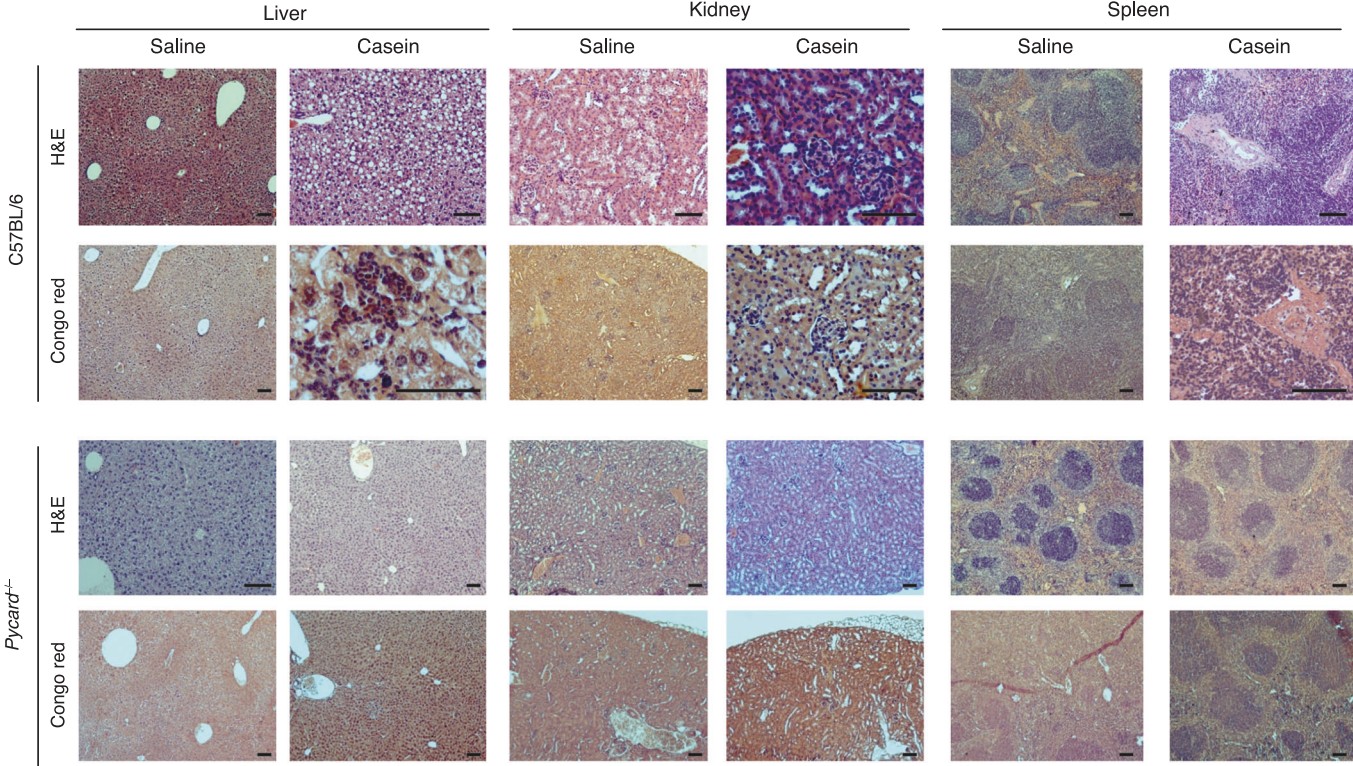

**Figure EV2.  Amyloidosis is reduced in ASC-deficient mice.**

Congo-red and Haematoxylin and eosin stained in liver, kidney and spleen sections from C57BL/6 and *Pycard⁻/⁻* mice after 25 days of casein administration. Please note that part of these images are also presented in Fig. 1F. Scale bars: 100 μm.

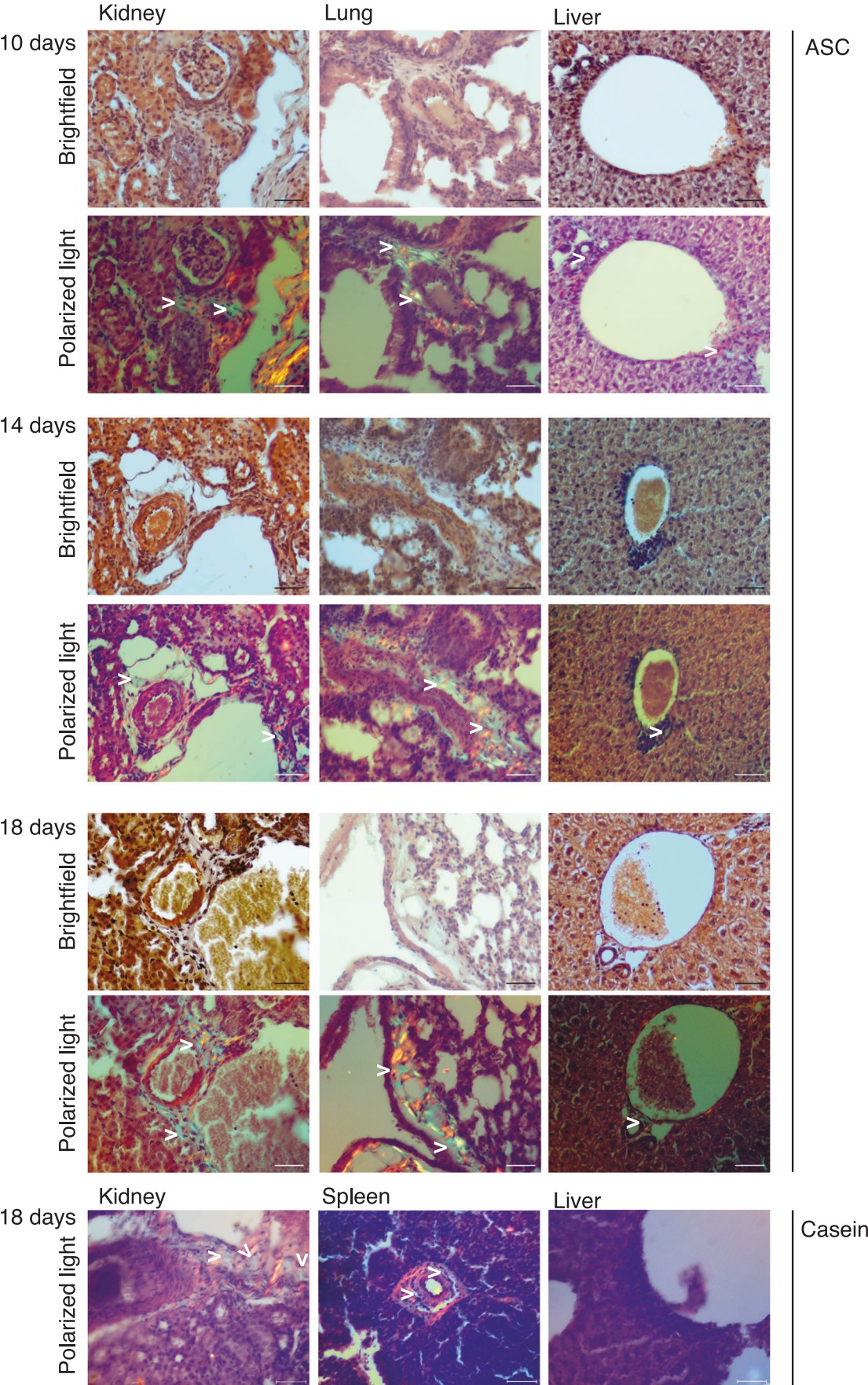

**Figure EV3.  Administration of ASC oligomers induce amyloidosis in vivo.**

Congo-red stained in kidney and lung sections from *Pycard*<sup>-/-</sup> mice after 10, 14 and 18 days of intraperitoneal ASC oligomers administration. Pictures of brightfield and polarized light are shown. As control, kidney, spleen and liver sections from C57BL/6 mice after 18 days of casein administration were stained with Congo-red and visualized under polarized light. Arrowheads denote areas of amyloid deposition. Please note that part of these images are also presented in Fig. 1J. Scale bars: 50 μm.

