## [Peer Review File · EMBO Molecular Medicine]

The inflammasome protein ASC promotes amyloid deposition in Cryopyrin-associated Periodic Syndromes

Cristina Alarcón-Vila, Laura Hurtado-Navarro, Sandra Mateo, Alejandro Peñín-Franch, Carlos Martínez, Cristina Molina-Lopez, María Baños, Ana Gómez, Javier Gomez-Roman, Alberto Baroja-Mazo, Juan Arostegui, Natalia Palmou-Fontana, Juan Martínez-García, and Pablo Pelegrin

Corresponding author: Pablo Pelegrin (pablo.pelegrin@imib.es)

Review Timeline:

Submission Date:	23rd Sep 24
Editorial Decision:	8th Oct 24
Revision Received:	6th Nov 24
Accepted:	7th Nov 24

Editor: Zeljko Durdevic

Transaction Report:

8th Oct 2024

Dear Dr. Pelegrin,

Thank you for the submission of your manuscript to EMBO Molecular Medicine. I am pleased to inform you that we will be able to accept your manuscript pending the following final amendments:

- 1) Please implement all referee suggestions.
- 2) Authors: We note following name discrepancies: Ana I Gomez and José J Gómez-Román in the manuscript while Ana Gómez-Sánchez and Javier Gomez-Roman in our submission system. Please correct.
- 3) Figures: Please upload the Extended View Figure 1 and 2 as separate, high-resolution files and move their legends to the end of the main manuscript file. Rename the figures to Figure EV1 and EV2 and updated their callouts in the main text. Also, please indicate in the Figure EV1 and EV2 legends that the images are also presented in the Figure 1F and 1J, respectively.
- 4) Author checklist: Please submit a complete checklist. <https://www.embopress.org/pb-assets/embosite/EMBO%20Press%20Author%20Checklist-1642513524327.xlsx>
- 5) In the main manuscript file, please do the following:
 - Please address all comments suggested by our data editors listed below:
 - o Figure legends:
 1. Please note that the legend for figure 1 is not labelled in the manuscript, however the corresponding figure is labelled as 1. This needs to be rectified.
 2. Please note that the figure title for figures EV 1, EV 2 is missing in the manuscript. This needs to be rectified.
 3. Please define the annotated p values ****/**/* as well as provide the exact p-values for the same in the legend of figure 1d-e; as appropriate.
 4. Please note that the exact p values are not provided in the legend of figure 1g.
 5. Please indicate the statistical test used for data analysis in the legends of figures 1d-e, g.
 6. Please note that information related to n is missing in the legends of figures 1d-e, g-i.
 7. Please note that the error bars are not defined in the legends of figures 1d-e, g-i.
 8. Please note that the scale bar needs to be defined for figure EV 2.
 9. Please note that scale bar and its definition are missing for figure EV 1.
 10. Please note that the open arrows are not defined in the legend of figure EV 2. This needs to be rectified.
 - Please rename "Dataset" to "Dataset EV1" and add callout(s) in the main manuscript text.
 - Rename "Material and methods" to "Methods" and move it to the manuscript file after the main text.
 - Please rename "Data sharing" to "Data availability" and place it after "Methods". Please replace the current sentence to "This study includes no data deposited in external repositories".
 - Rename "Funding" to "Acknowledgments" and move it to the main manuscript text before references.
 - Rename "Competing interests" to "Disclosure Statement & Competing Interests" and place it before "Acknowledgements". We updated our journal's competing interests policy in January 2022 and request authors to consider both actual and perceived competing interests. Please review the policy <https://www.embopress.org/competing-interests> and update your competing interests if necessary.
 - Remove "Patient and Public Involvement statement".
 - Author contributions: Please remove it from the manuscript. CRediT has replaced the traditional author contributions section because it offers a systematic machine-readable author contributions format that allows for more effective research assessment. You are encouraged to use the free text boxes beneath each contributing author's name to add specific details on the author's contribution. More information is available in our guide to authors: <https://www.embopress.org/page/journal/17574684/authorguide#authorshipguidelines>
 - In Methods, please include statement that the experiments with patient samples conformed to the principles set out in the WMA Declaration of Helsinki and the Department of Health and Human Services Belmont Report.
 - Please include structured Methods section that includes a Reagents and Tools Table (should be uploaded as a separate file) followed by a Methods and Protocols section. More information on how to adhere to this format as well as downloadable templates (.docx) for the Reagents and Tools Table can be found in our author guidelines: <https://www.embopress.org/page/journal/17574684/authorguide#structuredmethods>
 - An example of a paper with Structured Methods can be found here: <https://www.embopress.org/doi/full/10.1038/s44320-024-00037-6#sec-4>
- 6) Please include one (two) sentence summary of your findings in the point-by-point response.
- 7) As part of the EMBO Publications transparent editorial process initiative (see our Editorial at <http://embomolmed.embopress.org/content/2/9/329>), EMBO Molecular Medicine will publish online a Review Process File (RPF) to accompany accepted manuscripts. This file will be published in conjunction with your paper and will include the anonymous referee reports, your point-by-point response and all pertinent correspondence relating to the manuscript. Let us know whether you agree with the publication of the RPF and as here, if you want to remove or not any figures from it prior to publication. Please note that the Authors checklist will be published at the end of the RPF.
- 8) Please provide a point-by-point letter INCLUDING my comments as well as the reviewer's reports and your detailed responses (as Word file).

I look forward to reading a new revised version of your manuscript as soon as possible.

Yours sincerely,

Zeljko Durdevic

*** Instructions to submit your revised manuscript ***

- 1) a .docx formatted version of the manuscript text (including Figure legends and tables)
- 2) Separate figure files*
- 3) supplemental information as Expanded View and/or Appendix. Please carefully check the authors guidelines for formatting Expanded view and Appendix figures and tables at <https://www.embopress.org/page/journal/17574684/authorguide#expandedview>
- 4) a letter INCLUDING the reviewer's reports and your detailed responses to their comments (as Word file).
- 5) EMBO Molecular Medicine now requires a complete author checklist (<https://www.embopress.org/page/journal/17574684/authorguide>) to be submitted with all revised manuscripts. Please use the checklist as guideline for the sort of information we need WITHIN the manuscript. The checklist should only be filled with page numbers where the information can be found. This is particularly important for animal reporting, antibody dilutions (missing) and exact values and n that should be indicated instead of a range.
- 6) A Conflict of Interest statement should be provided in the main text
- 7) Please note that we now mandate that all corresponding authors list an ORCID digital identifier. This takes <90 seconds to complete. We encourage all authors to supply an ORCID identifier, which will be linked to their name for unambiguous name identification.

Currently, our records indicate that the ORCID for your account is 0000-0002-9688-1804.

Please click the link below to modify this ORCID:
Link Not Available

- 8) Include a Reagents and Tools Table as part of the Methods section, which can be downloaded from our author guidelines (<https://www.embopress.org/page/journal/17574684/authorguide#structuredmethods>)

Photos 400-800 DPI

*Additional important information regarding figures and illustrations can be found at

<https://bit.ly/EMBOPressFigurePreparationGuideline>. See also figure legend preparation guidelines:

<https://www.embopress.org/page/journal/17574684/authorguide#figureformat>

***** Reviewer's comments *****

Referee #1 (Novelty/Model system Comments for Author):

The models are appropriate.

Referee #1 (Remarks for Author):

Dear Dr. Durdevic,

I have carefully reviewed the manuscript titled "ASC promotes amyloidosis in Cryopyrin-Associated Periodic Syndromes (CAPS)."

Overall, the study is novel and provides a significant contribution to the understanding of ASC speck's role in amyloid deposition in CAPS. The findings of this manuscript complement those of Losa et al. (2024), who identified ASC as a key player in serum amyloid deposition. The focus on CAPS in the current manuscript is unique and adds specificity to the broader discussions presented in the other studies.

The manuscript provides strong clinical relevance by integrating a case study of a CAPS patient with a somatic NLRP3 variant alongside experimental models. This combination offers a compelling clinical and mechanistic foundation for understanding ASC's role in amyloid deposition. The authors further enhance the study's impact through a comprehensive methodology, employing a robust combination of human data, in vitro cell culture experiments, and in vivo mouse models. This approach provides a holistic view of ASC's involvement in amyloidosis, making the findings both thorough and well-substantiated. Additionally, the discussion of ASC-targeted therapies, particularly the potential use of nanobodies, positions the work within the realm of translational research, highlighting its significant clinical potential for future therapeutic applications.

Given the novelty, the strong experimental design, and the clinical implications, I recommend the manuscript for publication with minor revisions.

From a reader perspective, I would not start the introduction with "The article by Losa et al, (2024) reports that extracellular deposition of the inflammasome adaptor protein ASC in tissues leads to amyloid A (AA) amyloidosis." Instead, I would provide one or two sentences explaining the broader concept of amyloidosis (like the very next sentence) followed by a brief (one sentence) description of inflammasomes and how they're involved in AA, etc..

It is interesting to observe that the refereed patient has a myeloid-restricted variant c.924A>T. Am I right to interpret that these findings would indicate that myeloid cells would be significant sources of extracellular ASC? In the case of the mouse model for example, would the main cells producing/releasing ASC specks be myeloid innate immune cells? I suppose this could be easily addressed using Flow Cytometry (or ImageStream) with specific surface markers. There are well-established protocols to assess cells with ASC specks by FACS.

In Figure 1D and/or E, I would recommend (if space allows it) to include representative confocal images that support the quantification shown as a graph.

Point-by-point rebuttal letter

Title: The inflammasome adaptor protein ASC promotes amyloid deposition in Cryopyrin-associated Periodic Syndromes

MS No. EMM-2024-20633

To Dr. Zeljko Durdevic, Editor EMBO Molecular Medicine,

General: We are especially grateful to the reviewer for his/her constructive comments that have greatly improved the quality of the manuscript. In this revision, we have revised the text and figures in accordance with all the reviewer suggestions. All their comments have been addressed in the corrected version of our manuscript and in the point-by-point rebuttal letter. Changes in the MS text are highlighted in yellow in the marked copy. Also, a clean copy is provided with the submission.

In this study we found that the deposition of amyloid in tissues in Cryopyrin-Associated Periodic Syndrome (CAPS) were promoted by the extracellular presence of the inflammasome adaptor protein ASC, opening exciting new directions in clinical practice to obtain a novel therapy towards secondary amyloidosis in inflammasomopathies.

We hope you will find the revised MS suitable for publication in EMBO Molecular Medicine.

Reviewer's comments

Referee #1 (Novelty/Model system Comments for Author): The models are appropriate.

Answer: we thank the reviewer to find our models appropriate for the present study.

Referee #1 (Remarks for Author):

Dear Dr. Durdevic,

I have carefully reviewed the manuscript titled "ASC promotes amyloidosis in Cryopyrin-Associated Periodic Syndromes (CAPS)."

Overall, the study is novel and provides a significant contribution to the understanding of ASC speck's role in amyloid deposition in CAPS. The findings of this manuscript complement those of Losa et al. (2024), who identified ASC as a key player in serum amyloid deposition. The focus on CAPS in the current manuscript is unique and adds specificity to the broader discussions presented in the other studies.

The manuscript provides strong clinical relevance by integrating a case study of a CAPS patient with a somatic NLRP3 variant alongside experimental models. This combination offers a compelling clinical and mechanistic foundation for understanding ASC's role in amyloid deposition. The authors further enhance the study's impact through a comprehensive methodology, employing a robust combination of human data, in vitro cell culture experiments, and in vivo mouse models. This approach provides a holistic view of ASC's involvement in amyloidosis, making the findings both thorough and well-substantiated. Additionally, the discussion of ASC-targeted therapies, particularly the potential use of nanobodies, positions the work within the realm of translational research, highlighting its significant clinical potential for future therapeutic applications.

Given the novelty, the strong experimental design, and the clinical implications, I recommend the manuscript for publication with minor revisions.

Answer: We appreciate reviewer comment about the novelty and clinical relevance of our study, that covers patient data, animal models and in vitro experiments, to show that extracellular ASC oligomers in CAPS patients could be foci of amyloid deposition and initiating amyloidosis.

From a reader perspective, I would not start the introduction with "The article by Losa et al, (2024) reports that extracellular deposition of the inflammasome adaptor protein ASC in

tissues leads to amyloid A (AA) amyloidosis." Instead, I would provide one or two sentences explaining the broader concept of amyloidosis (like the very next sentence) followed by a brief (one sentence) description of inflammasomes and how they're involved in AA, etc..

Answer: As suggested by the reviewer, we have now revised the text and changed the initial paragraph of the MS.

It is interesting to observe that the refereed patient has a myeloid-restricted variant c.924A>T. Am I right to interpret that these findings would indicate that myeloid cells would be significant sources of extracellular ASC? In the case of the mouse model for example, would the main cells producing/releasing ASC specks be myeloid innate immune cells? I suppose this could be easily addressed using Flow Cytometry (or ImageStream) with specific surface markers. There are well-established protocols to assess cells with ASC specks by FACS.

Answer: The reviewer is right to interpret that myeloid cells are causative of CAPS pathology and a source of extracellular ASC, this is supported by other studies reporting patients with late on-set CAPS with myeloid restricted NLRP3 variants (i.e. PMID: 27273849, 29163488), these patients present elevation in the serum of SAA and ASC oligomers, developing AA amyloidosis in kidney (PMID: 29163488). In these patients anti-IL-1 therapy decreased extracellular ASC oligomers, SAA concentration and improved renal function. Furthermore, mouse models expressing NLRP3-associated CAPS variants in macrophages were enough to induce autoinflammation (PMID: 35574994).

The identification of the endogenous cells releasing ASC in CAPS is an interesting issue. However, we have analyzed blood sample of many different CAPS patients with germline/somatic mutations, and we were unable to find any blood cell with already formed ASC after flow cytometry analysis. Maybe ASC specking cells are dying in vivo and could not be detected ex vivo. We now show in the revised MS that after 4 h in culture, blood monocytes (and not granulocytes or lymphocytes) were producing ASC oligomers. This aligns with the expression of ASC in these cells, as we found that monocytes expressed more ASC than neutrophils (please see new data on Figure 1F and EV1C). Unfortunately, we could not analyze blood sample from the reported late-onset CAPS patient, as it deceased in June 2024. We have now included this new data in the revised MS showing that in CAPS monocytes are the cells producing ASC specks (and probably the cells releasing them) and enriched the discussion in this interesting point including the references above mentioned.

In Figure 1D and/or E, I would recommend (if space allows it) to include representative confocal images that support the quantification shown as a graph.

Answer: As requested by the reviewer, representative images to show NLRP3 puncta distribution in HEK293 cells transfected with either NLRP3 wild type or p.Q308H tagged with YFP are shown in the new Figure EV1A. These preps correspond to the quantified data shown in Figure 1D. For Figure 1E, we did not have pictures, as these cells were directly analyzed by flow cytometry. For this flow cytometry we gated cells with a low expression of ASC (low ASC-RFP MFI) and increasing expression of NLRP3 (increasing NLRP3-YFP MFI), as shown in the new Figure EV1B. Therefore, microscopy of these transfections will not support the data presented in Figure 1E, as you can see, the analyzed cells are a proportion of all the transfected cells.

7th Nov 2024

Dear Dr. Pelegrin,

We are pleased to inform you that your manuscript is accepted for publication and is now being sent to our publisher to be included in the next available issue of EMBO Molecular Medicine.
